# FORESIGHTKV: OPTIMIZING KV CACHE EVICTION FOR REASONING MODELS BY LEARNING LONG-TERM CONTRIBUTION

## ABSTRACT

Recently, large language models (LLMs) have shown remarkable reasoning abilities by producing long reasoning traces. However, as the sequence length grows, the key-value (KV) cache expands linearly, incurring significant memory and computation costs. Existing cache eviction methods mitigate this issue by discarding less important KV pairs, but often fail to capture complex KV dependencies, resulting in performance degradation. To better balance efficiency and performance, we introduce ForesightKV, a training-based KV cache eviction framework that learns to predict which KV pairs to evict during long-text generations. We first design the Golden Eviction algorithm, which identifies the optimal eviction KV pairs at each step using future attention scores. These traces and the scores at each step are then distilled via supervised training with a Pairwise Ranking Loss. Furthermore, we formulate cache eviction as a Markov Decision Process and apply the GRPO algorithm to mitigate the significant language modeling loss increase on low-entropy tokens. Experiments on AIME2024 and AIME2025 benchmarks with Qwen3-1.7B and Qwen3-4B demonstrate that ForesightKV consistently outperforms prior methods under only half the cache budget, while benefiting synergistically from both supervised and reinforcement learning approaches.

## 1 INTRODUCTION

Recently, large language models (LLMs), especially the reasoning models, have demonstrated exceptional long-context generation capacities (Comanici et al., 2025; DeepSeek-AI et al., 2025; Zhao et al., 2023; Chen et al., 2025). With the extension of context windows and advancements in large-scale reinforcement learning, these models can generate complex and long reasoning paths for reasoning tasks, enabling them to solve problems through step-by-step reasoning (DeepSeek-AI et al., 2025; Shao et al., 2024; Peng et al., 2024). However, as the model's output length increases, the size of the model's internal key-value (KV) cache grows linearly, leading to decreased generation speed and increased memory overhead. Taking Qwen3-4B (Yang et al., 2025a) as an example, at a sequence length of 32K, the KV cache storage for a single instance consumes 4.5 GB with the precision of BFloat16, severely limiting the number of concurrent batches. Furthermore, since decoding is an inherently memory-bound process, loading an excessively long KV cache introduces significant latency (Yuan et al., 2024).

To address the overhead of the KV cache in long texts, numerous KV cache compression methods have been proposed, which reduce costs by either permanently reducing the number of KV pairs or reducing the overhead of storing a single KV pair (Li et al., 2024; Zhang et al., 2023; Liu et al., 2024; Wang et al., 2024; Chang et al., 2024). Among them, the majority are designed for long-input tasks, with only a few studies being applicable to long-text generation. During the generation process, these methods typically employ elaborately designed rules (*e.g.,* attention scores, features of the KV pairs, and their positions) to estimate the importance of KV pairs and permanently evict unimportant ones at each eviction step (Cai et al., 2025; Zhang et al., 2023; Xiao et al., 2024; Wu et al., 2024). However, these training-free methods are often insufficient to take the all complex patterns across different attention heads into account, typically leading to suboptimal performance. In contrast, other works employ training-based methods to perform a one-time evaluation of KV pair

importance for eviction (Lancucki et al., 2025), which fails to capture the dynamic nature of their importance at different stages of the sequence.

To achieve a better tradeoff between generation quality and efficiency in the KV cache eviction process, we first investigate the behavior of Qwen3-4B (Yang et al., 2025a) on questions and reasoning traces generated by itself. We observe that KV pairs across different attention heads often exhibit different patterns, which can be categorized into three main types, *i.e.,* global, position-dependent, and semantic-dependent (as shown in Figure 1). Among them, due to the inherent properties of reasoning data, semantic-dependent patterns exhibit greater complexity compared to the other two types, manifesting features such as block-wise attention structures and dynamic variations. This makes it difficult to capture them with rule-based algorithms or methods that determine importance in a single pass (Zhang et al., 2023; Lancucki et al., 2025). In addition, we observe that the losses of low-entropy tokens sharply change due to the absence of critical KV pairs, resulting in factual errors that may distort subsequent reasoning. These observations highlight the necessity of eviction methods that can adaptively track semantic dependencies and mitigate the significant loss increase of low-entropy tokens during generation.

In this work, we propose ForesightKV, a method that learns to predict the long-term contribution of each KV pair and adaptively evicts less important ones during generation. The key idea is to train a lightweight scoring model that estimates the dynamic importance of each KV pair and guides eviction decisions. To achieve this, ForesightKV employs a two-stage training paradigm: supervised learning with constructed future importance labels, followed by reinforcement learning to refine eviction policies.We first introduce the *Golden Eviction* algorithm to construct supervision labels. Specifically, we partition the attention matrix of a sequence into fixed-length blocks along the query dimension and aggregate attention scores within each block and attention heads within the same group to compute block scores. For each eviction step, we employ the maximum block scores of future steps as future scores and discard pairs with the lowest future scores, thus minimizing the impact of eviction. The scoring model is then trained with these labels using a pairwise ranking loss to capture relative eviction preferences. Subsequently, we model the KV cache eviction as a Markov Decision Process (MDP) and optimize the scoring models with reinforcement learning. On a full reasoning sequence, the scoring model selects eviction actions at each step, which results in different eviction traces. We formulate the sequence reward based on the MSE loss of language modeling losses before and after KV cache eviction for tokens that are initially predicted with low entropy and with large loss increases. The scoring models are then trained using the GRPO algorithm to maximize this cumulative reward. Notably, our framework trains solely the lightweight scoring models with no backward operations or parameter updates on the LLMs.

To evaluate the effectiveness of our method, we compare ForesightKV with other KV cache eviction methods on two reasoning LLMs using two math benchmarks, *i.e.,* AIME2024 and AIME2025. Experiments show that our method can achieve better results than the baselines with even only half the KV cache. Specifically, our method retains $90\%$ and $97\%$ of the original model performance under 2K and 4K budgets, respectively, striking a balance between performance and efficiency.

## 2 EMPIRICAL STUDY

### 2.1 COMPLEX ATTENTION PATTERNS DURING LONG-TEXT REASONING

To investigate the properties of the KV cache during long-context reasoning, we employ Qwen3-4B (Yang et al., 2025a) to generate responses to questions from the STILL dataset (Min et al., 2024). We then select a representative response to calculate attention scores and visualize the attention maps of three attention heads, as illustrated in Figure 1. Consistent with the findings of MInference 1.0 (Jiang et al., 2024), we observe distinct attention patterns across different attention heads. Based on the dynamic characteristics of these attention patterns, we categorize the KV pairs into three types, namely global, position-dependent, and semantic-dependent.

- *Global*: Global KV pairs commonly manifest as vertical lines in the attention map. This visual pattern indicates that these pairs receive a high attention weight from the vast majority of queries.

- *Position-dependent*: In Transformer decoders, the attention mechanism often demonstrates locality, with most of the attention allocated to tokens near the query (Xiao et al., 2024). As the decoding length increases, attention to earlier KV pairs decreases, reducing their overall importance.

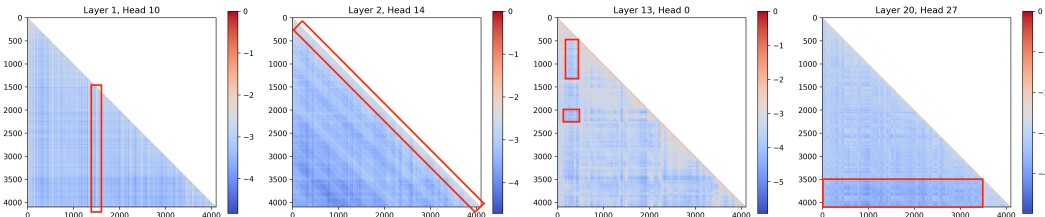

Figure 1: KV patterns in Qwen3-4B, which can be divided into the following patterns: (A) Global; (B) Position-dependent; (C) and (D) Semantic-dependent.

• *Semantic-dependent*: In contrast to the other two types, semantic-dependent KV pairs exhibit more complex properties. As shown in Figure 1 (C), the attention map often displays a block-wise pattern due to inherent structural features in long-text reasoning data. In this case, attention is primarily concentrated on KV pairs within the current block and in previously semantically related blocks. As decoding moves into subsequent blocks, the high-attention regions may shift accordingly. Moreover, after a specific query, KV pairs that previously received high attention may become permanently irrelevant, as illustrated in Figure 1 (D).

Given the complexity of these attention mechanisms, previous KV cache eviction algorithms often fail to capture such intricate patterns. For instance, for each eviction step, SnapKV (Li et al., 2024) uses the last few tokens as an observation window and utilizes their attention scores to evaluate the importance of KV pairs. However, many semantic-dependent KV pairs will often exhibit different importance as the semantics change. Therefore, SnapKV often discards tokens that have little semantic association with the tokens of that window but are subsequently important, causing severe performance loss. Furthermore, these patterns are not mutually exclusive and can co-occur within a single attention head, posing an additional challenge for KV cache eviction design.

## 2.2 LARGE SHIFTS IN LOSS FOR LOW-ENTROPY TOKENS

In reasoning tasks, the entropy of LLMs plays a crucial role. High-entropy tokens (tokens with top-20% entropy) often mark decision points between different reasoning paths, while low-entropy tokens (other tokens) make up the more detailed and deterministic parts of the reasoning process (Wang et al., 2025). To investigate how these distinct token types behave, we employ Qwen3-4B (Yang et al., 2025a) to compute the loss on given sequences and compare the changes in loss for high-entropy and low-entropy tokens with R-KV (Cai et al., 2025) across math, coding and summarization tasks. As shown in Table 1, low-entropy tokens are influenced largely by the KV cache eviction, which is reflected as a greater proportional increase in the loss. We infer that this may be due to the eviction of contextual information, which is crucial for maintaining the low entropy of these tokens (Qiu et al., 2025). Furthermore, we analyze prediction errors on low-entropy tokens and find they often involve numbers and symbols (see Appendix C), whose factual mistakes can accumulate over long sequences and distort subsequent reasoning. Thus, it is very important to consider these low-entropy tokens for KV cache eviction.

Table 1: Loss ratio of tokens with different entropies after eviction.

| Tokens | Math | Code | Sum |
|--------|------|------|-----|
| Low | +147% | +75% | +187% |
| High | +52% | +1% | +142% |

## 3 FORESIGHTKV

To effectively manage the KV cache in long-context generation, we introduce **ForesightKV**, a training-based eviction method that balances memory efficiency with generation quality. The key idea is to learn a scoring model with foresight to predict the long-term contribution of KV pairs and guide eviction decisions under a fixed memory budget. Specifically, our approach begins by formally defining the compression process during long-context generation with scoring models in our framework (Section 3.1). We then propose a two-stage training pipeline consisting of supervised

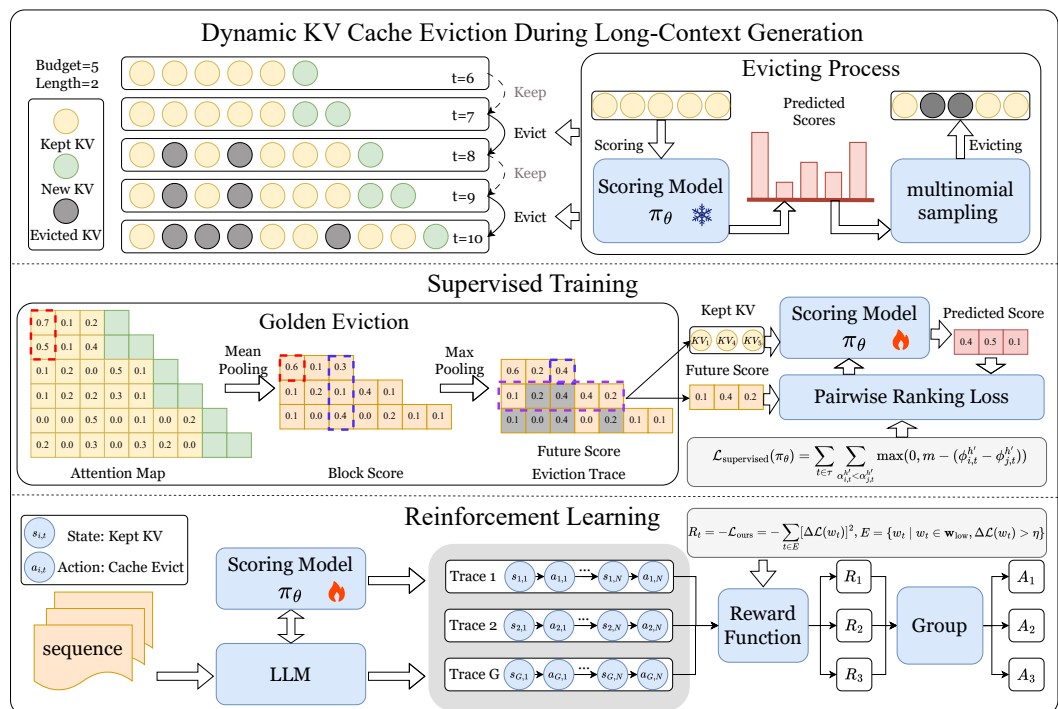

Figure 2: Overview of ForesightKV. ForesightKV uses a scoring model to guide dynamic KV cache eviction during long-context generations, which is trained through supervised learning and then reinforcement learning. The dashed box indicates pooling.

training (Section 3.2) and reinforcement learning (Section 3.3) to optimize the scoring models' long-term prediction capacities. A flowchart of our method is presented in Figure 2.

## 3.1 DYNAMIC KV CACHE EVICTION DURING LONG-CONTEXT GENERATION

In general, we define a dynamic compression process in our framework, parameterized by two hyperparameters: a cache budget $B$, which specifies the target number of KV pairs to retain, and an eviction length $L$, which determines the frequency of the eviction process. The eviction is triggered periodically: after every $L$ new tokens are generated, the cache size reaches $B + L$. At this point, we keep the most recent $L$ KV pairs and employ an eviction algorithm that prunes the other cache back to the budget size $B - L$ by retaining only the most important KV pairs. Specifically, at each eviction step, we assign a scoring model $\pi_\theta$ to evaluate the importance of each KV pair. To balance efficiency and performance, we adopt an MLP scorer $\pi_\theta$ to predict the importance score $\phi_n$ of the $n$-th KV pair. Motivated by the complex and diverse KV cache patterns observed in Section 2.1, we construct the input feature $\mathbf{x}_n$ by concatenating the key $\mathbf{k}_n$, the value $\mathbf{v}_n$, and their associated attention features $\mathbf{a}_n$, fixed-length vectors transformed from the attention scores (see Appendix B).

$$\mathbf{x}_n = \text{Concat}(\mathbf{k}_n, \mathbf{v}_n, \mathbf{a}_n), \quad \phi_n = \pi_\theta(\mathbf{x}_n). \tag{1}$$

Based on the importance scores, we first select $2L$ KV pairs via Top-K sampling, and then select $L$ pairs from them via multinomial sampling. We drop the $L$ less important KV pairs and retain the rest for subsequent attention computation:

$$s'_t = s_t / \text{Multinomial-}L((\text{Softmax}(\text{Top-}2L(\Phi))). \tag{2}$$

We provide the details of the scoring model and sampling method in Appendix B. In the following, we present the two-stage training framework for the scoring models. In the first stage, we employ the "Golden Eviction" algorithm that leverages future information to generate optimal eviction traces, which serve as supervision for training the scoring model. In the second stage, we refine the model via reinforcement learning by formulating eviction as a Markov Decision Process. Throughout training, the LLM parameters remain frozen, and only the lightweight scoring models are updated.

## 3.2 SUPERVISED TRAINING

In our two-stage framework, we begin with a supervised training phase to equip the scoring model with the ability to identify critical KV pairs. This stage is motivated by our analysis (Section 2), which shows that the relevance of a KV pair evolves dynamically during generation. To predict the future importance, we introduce the *Golden Eviction* algorithm, which constructs target labels and eviction traces. The scoring models are then trained on these labels using a Pairwise Ranking Loss.

**Golden Eviction.** To make the computation of target labels more tractable, we aggregate the attention dimensions according to the eviction steps and the group size in GQA. Specifically, for a given full reasoning trace consisting of the prompt and generated response, $\mathbf{w} = \{w_1, \ldots, w_T\}$, we compute the corresponding attention score matrix $\mathbf{A}^h \in \mathbb{R}^{T \times T}$ on the original model, where $h$ denotes head indice on given layer. Subsequently, we partition the attention matrix along the query dimension with a stride of $L$, starting from the first eviction position $B + L$, where the final block is padded to match the maximum length. We employ pooling across the query dimension for each block to obtain the block score $\mathbf{a}_t^h$ for each kv pair:

$$\mathbf{a}_t^h = \text{Pooling}(\mathbf{A}^h[B + tL : B + (t+1)L - 1, :]) \in \mathbb{R}^T, \ t \in \{1, \ldots, M\}, \tag{3}$$

where $M = \lceil (T - B)/L \rceil - 1$ is the number of eviction steps. Specifically, for GQA, where $g$ attention heads share the same KV cache, we also employ pooling of these scores within one attention group $h'$ to make a unified eviction for each group. For simplicity, we employ average pooling for the two pooling computations:

$$\tilde{\mathbf{a}}_t^{h'} = \text{Pooling}(\{\mathbf{a}_t^{g(h'-1)+1}, \ldots, \mathbf{a}_t^{gh'}\}) \in \mathbb{R}^T. \tag{4}$$

For each eviction step $t$, our objective is to minimize the impact on future attention computations. In other words, we want the evicted KV pairs to have low attention scores in all future blocks. Thus, we first compute the maximum block score of all future blocks as the future score $\alpha_{i,t}^{h'} \in \mathbb{R}$ for the $i$-th KV pair and only keep KV pairs with the largest future scores as the kept cache $s_t^{h'}$.

$$\alpha_{i,t}^{h'} = \max_{t \le j \le M}(\tilde{\mathbf{a}}_{i,t}^{h'}), \quad s_t^{h'} = \text{Top-}(B - L)(\alpha_{i,t}^{h'}). \tag{5}$$

**Training.** Subsequently, based on the KV cache eviction traces obtained from the Golden Eviction algorithm, we employ the scoring model to compute a score for each KV pair at every eviction step. Furthermore, we frame the eviction as a ranking task and use Pairwise Ranking Loss to train the scoring models. Our objective is to ensure that the model's predicted scores for any two KV pairs are ranked in the opposite order of their future scores. For any pair of indices $i$ and $j$, if $\alpha_{i,t}^{h'} < \alpha_{j,t}^{h'}$, we require that $\phi_{i,t}^{h'} > \phi_{j,t}^{h'}$. Formally, we define the training loss function as follows ($m$ is a hyper-parameter for the loss):

$$\mathcal{L}_{\text{supervised}}(\pi_\theta) = \sum_{t \in \tau} \sum_{\alpha_{i,t}^{h'} < \alpha_{j,t}^{h'}} \max(0, m - (\phi_{i,t}^{h'} - \phi_{j,t}^{h'})). \tag{6}$$

**Effectiveness of Golden Eviction.** To demonstrate the effectiveness of the Golden Eviction algorithm, we compare it against three leading KV cache eviction algorithms: R-KV (Cai et al., 2025), SnapKV Li et al. (2024), and H2O (Zhang et al., 2023). Using Qwen3-4B (Yang et al., 2025a) models, we measure the percentage increase in model loss on sampled inference data. We evaluate these methods with the KV cache budgets of 1024 and 2048, and employ the KV cache eviction every 128 or 256 tokens. As detailed in Table 2, Golden Eviction consistently yields significantly lower loss than the competing approaches. These results confirm the effectiveness of Golden Eviction in preserving model performance under strict memory constraints.

## 3.3 REINFORCEMENT LEARNING

For KV cache eviction in long-context decoding, the retained KV cache influences both the current generation of the model and subsequent cache eviction decisions. Therefore, we define KV cache

Table 2: Comparison of model loss ratios for different methods relative to the original model under various KV cache budgets. Here, (1024,128), (2048,128), (1024,256), and (2048,256) denote KV cache budgets of 1024 and 2048, with KV cache evicts every 128 or 256 tokens.

| Model | Method | (1024,256) | (2048,256) | (1024,128) | (2048,128) |
|---|---|---|---|---|---|
| Qwen3-4B | Golden | **1.0711** | **1.0166** | **1.0715** | **1.0185** |
| | R-KV | 1.4101 | 1.1606 | 1.4750 | 1.1814 |
| | SnapKV | 1.4091 | 1.1281 | 1.4214 | 1.1343 |
| | H2O | 1.2730 | 1.0948 | 1.4106 | 1.1578 |

eviction during inference as a Markov Decision Process (MDP) and employ reinforcement learning algorithms to optimize the eviction process. Given a full reasoning trace $\mathbf{w} = \{w_1, \ldots, w_T\}$, and the frozen LLM $\Theta$, our goal is to maximize the rewards by optimizing the set of scoring models $\{\pi_{\theta_{h,l}}\}$. In the following, we first define the components of the reinforcement learning framework:

• State ($s_t$): The current remaining KV cache at the step $t$.

• Action ($a_t$): Given a state $s_t$, the action $a_t$ is to select a subset of indices of the KV pairs $\{1, \ldots, B + L\}$ to retain for the current generation step by a predefined cache budget.

• Policy ($\pi_\theta$): For each attention group, we define the scoring model as the policy model, which assigns a score to guide whether the KV pair should be retained or discarded.

• Reward ($R_t$): To evaluate the quality of an eviction action, we define a sequence-level reward. Inspired by the observations in Section 2.2, we find that increases in the loss of low-entropy tokens have a substantial impact on reasoning quality. Based on this insight, we construct a subset of tokens $E$ according to two criteria: (1) the token's original entropy falls within the bottom $80\%$ of the sequence $\mathbf{w}_{\text{low}}$, and (2) its loss increase from the original model ($\mathcal{L}_{\text{ori}}$) to the evicted model ($\mathcal{L}_{\text{evict}}$) exceeds a threshold $\eta$, $\Delta\mathcal{L}(w_t) = \mathcal{L}_{\text{ori}}(w_t) - \mathcal{L}_{\text{evict}}(w_t)$. The subset $E$ is formulated as follows:

$$E = \{w_t \mid w_t \in \mathbf{w}_{\text{low}}, \Delta\mathcal{L}(w_t) > \eta\}. \tag{7}$$

The reward is then defined as the average square of the loss increases $\Delta\mathcal{L}(w_t)$ across all tokens in $E$, formulated as:

$$R_t = -\mathcal{L}_{\text{ours}} = -\sum_{t \in E}[\Delta\mathcal{L}(w_t)]^2. \tag{8}$$

We use the GRPO algorithm (Shao et al., 2024) to train the scoring models. We initialize our policy model $\pi_\theta$ with the model obtained from supervised training, which also serves as the reference model $\pi_{\text{ref}}$. At each training step, for a given sequence $\mathbf{w}$, we sample $G$ KV cache eviction traces for all layers and heads using the Top-K multinomial sampling method with the old policy $\pi_{\theta_{\text{old}}}$, $i.e.$, $\{o_1, \ldots, o_G\}$. Influenced by the different kept KV caches across traces, the same token will exhibit different hidden states, resulting in varying rewards throughout the entire sequence. Subsequently, we employ group relative normalization to compute the estimated advantage scores for these traces:

$$\hat{A}_t = (R_t - \text{Mean}(R_t))/\text{Std}(R_t). \tag{9}$$

Furthermore, we broadcast these advantage scores to every eviction process for all scoring models and optimize these models together using the following training objective.

$$\mathcal{J}(\theta) = \mathbb{E}_{o \sim \pi_{\theta_{\text{old}}}} \left[ \sum_{t=1}^{|o|} \min\left( \frac{\pi_\theta(a_t|s_t)}{\pi_{\theta_{\text{old}}}(a_t|s_t)} \hat{A}_t, \text{clip}(\frac{\pi_\theta(a_t|s_t)}{\pi_{\theta_{\text{old}}}(a_t|s_t)}, 1 - \epsilon, 1 + \epsilon)\hat{A}_t \right) \right.$$

$$\left. - \beta \cdot \text{KL}\left[\pi_\theta(a_t|s_t) \| \pi_{\text{ref}}(a_t|s_t)\right] \right], \tag{10}$$

where $\frac{\pi_\theta(a_t|s_t)}{\pi_{\theta_{\text{old}}}(a_t|s_t)}$ is the importance sampling ratio, $\epsilon \in \mathbb{R}$ is the clipping threshold, and $\beta$ controls KL regularization.

# 4 EXPERIMENTS

## 4.1 EXPERIMENTS SETUP

**Evaluated Models and Benchmarks.** To evaluate the effectiveness of our method on the long-text reasoning task, we choose Qwen3-4B and Qwen3-1.7B (Yang et al., 2025a) as evaluated models and evaluate them on two complex math benchmarks, *i.e.,* AIME2024 and AIME2025. Following the settings of previous work (DeepSeek-AI et al., 2025; Yang et al., 2025a), we set the temperature of $0.6$, top-$k$ of 20, and top-$p$ of $0.95$. To ensure statistical reliability, we report the average scores of pass@1, where each benchmark is evaluated independently 32 times. For each method, we evaluate the performance under different cache budgets $B$, *e.g.,* 1024, 2048, and 4096. For both models, we set the immediate size of scoring models to 16. We also select the top 512 KV pairs first and sample 256 pairs from them. We introduce the details of training setups in Appendix D.1.

**Baselines.** We compare our method with four KV cache eviction methods, *i.e.,* SnapKV (Li et al., 2024), H2O (Zhang et al., 2023), and R-KV (Cai et al., 2025). Following the settings in R-KV, we compress the KV cache every $L$ steps for all these methods and apply SnapKV during the long-decoding phase, where it performs dynamic compression at each step by leveraging the attention within a fixed-size window. The details of these methods are provided in Appendix D.2.

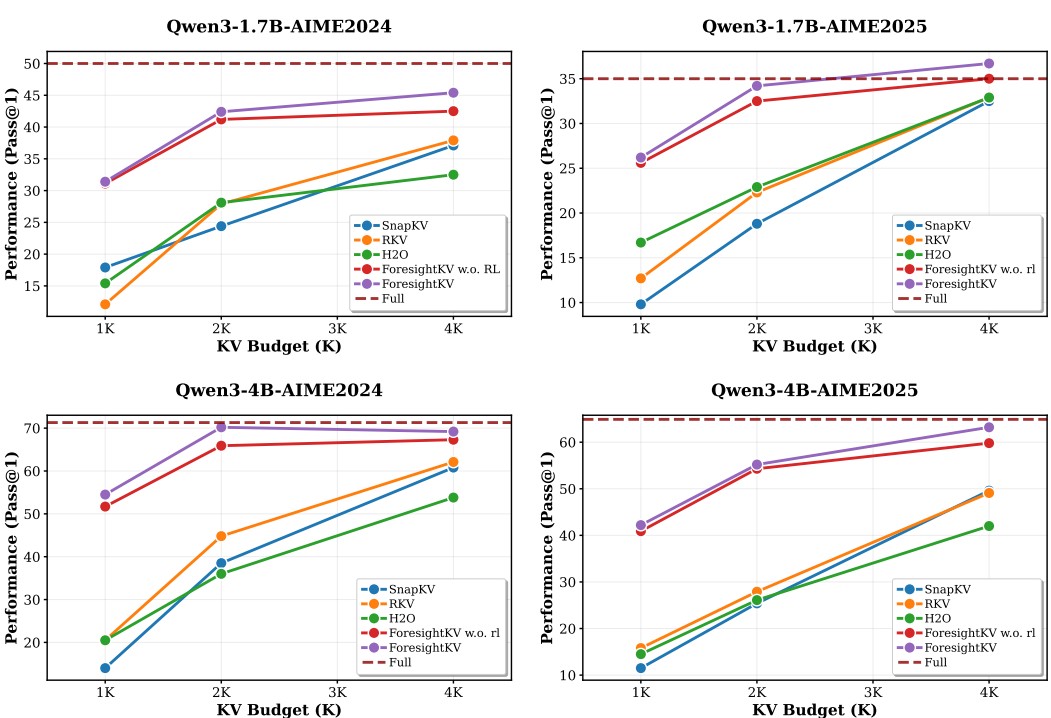

Figure 3: Comparison of ForesightKV with other KV cache eviction methods on reasoning tasks.

## 4.2 MAIN RESULTS

Figure 3 presents the performance comparison of ForesightKV and other KV cache eviction methods. Firstly, our method achieves better performance under the same KV cache budgets compared to other eviction methods across various benchmarks. Compared to other KV cache eviction methods, ForesightKV achieves comparable or even better performance with only half the KV cache. For example, on the AIME2024 dataset, ForesightKV with the Qwen3-4B model and a budget of only 1K outperforms R-KV with a 2K budget (54.5 vs. 44.8). Additionally, on certain benchmarks, ForesightKV can achieve comparable performances with the original model with a budget of 4K KV

pairs. This demonstrates that ForesightKV learns to better predict the long-term importance of KV pairs, thereby preserving the model's long-text reasoning capabilities more effectively.

Secondly, our method can benefit from both supervised training and reinforcement learning stages. The first stage of supervised training endows the scoring models with a powerful predictive capability for importance, significantly surpassing existing rule-based methods on benchmarks. Furthermore, the second stage of reinforcement learning enhances the scoring models by optimizing their capabilities through optimization of the overall LLMs. By optimizing the eviction policy to maximize the predictive accuracy of critical future tokens across all positions, the model's reasoning performance can be further improved within a fixed KV cache budget.

Finally, our method demonstrates generalization performance across different budgets. We can observe that although our method is trained with specific budgets in both training stages ($B \leq 2K$), it can naturally generalize to other budgets, *e.g.,* a budget size of 4K. This suggests that our training method enables the model to learn a generalizable understanding of the importance of different KV cache entries.

### 4.3 FURTHER ANALYSIS

**Reward Design.** To optimize the KV cache eviction, we design and evaluate several distinct reward functions. (1) $\mathcal{L}_{\text{all}}$, $\mathcal{L}_{\text{low}}$, and $\mathcal{L}_{\text{high}}$: the language modeling losses for all, low-entropy, and high-entropy tokens, respectively; (2) $\mathcal{L}_{\text{low,large}}$: the loss on the subset of low-entropy tokens whose loss increases by more than 1.5 post-eviction; and (3) $\mathcal{L}_{\text{ours}}$: the MSE loss on this same subset of severely impacted tokens as $\mathcal{L}_{\text{low,large}}$. We use the reinforcement learning framework on Qwen3-4B with the budget size of $1K$ and present the results in Table 3. Our findings reveal that naively minimizing the overall language modeling loss ($\mathcal{L}_{\text{all}}$) does not guarantee better performance. While targeting low-entropy tokens ($\mathcal{L}_{\text{low}}$) yields marginal gains, focusing on high-entropy ones ($\mathcal{L}_{\text{high}}$) causes significant degradation. The most effective strategy is to selectively optimize for low-entropy tokens that are most adversely affected by eviction. Using $\mathcal{L}_{\text{low,large}}$ and $\mathcal{L}_{\text{ours}}$ as reward signals reduces both the number of tokens with large loss spikes and the overall model perplexity. Notably, the MSE-based reward, $\mathcal{L}_{\text{ours}}$, is particularly effective at penalizing catastrophic loss increases, leading to the best final performance.

Table 3: Ablation study of reward function with 1K budget.

| Reward | AIME24 | AIME25 |
|---|---|---|
| - | 51.7 | 40.9 |
| -$\mathcal{L}_{\text{all}}$ | 50.6 | 40.0 |
| -$\mathcal{L}_{\text{low}}$ | 53.5 | 40.4 |
| -$\mathcal{L}_{\text{high}}$ | 49.6 | 35.4 |
| -$\mathcal{L}_{\text{low,large}}$ | 53.8 | **42.3** |
| -$\mathcal{L}_{\text{ours}}$ | **54.5** | **42.3** |

Table 4: Ablation study of scoring model and sampling function with 1K budget. *Attn* and *KV* denote using attention features and KV representations. *Top-K* and *MN* denote the Top-K and multinomial sampling methods.

| Input | Sampling | AIME24 | AIME25 |
|---|---|---|---|
| Attn+KV | Top-K+MN | **51.7** | **40.9** |
| Attn | Top-K+MN | 37.5 | 22.9 |
| Attn+KV | MN | 16.5 | 13.8 |
| Attn+KV | Top-K | 46.0 | 37.7 |

**Design of Model and Sampling Algorithm.** We conduct ablation studies to analyze the effects of the scoring model's inputs and the sampling algorithms. Instead of concatenating attention features with KV representations, we evaluate the scoring model with only attention features. We also replace our sampling method (multinomial sampling from the top-K pairs) with either top-K or multinomial sampling. As shown in Table 4, evaluations on AIME2024 and AIME2025 with Qwen3-4B reveal that both modifications consistently degrade performance: top-K and multinomial sampling led to a significant performance drop, while only employing attention features impaired eviction quality, indicating that the scoring model struggles to accurately predict the importance of KV pairs.

**Generalization Capacities.** To evaluate the generalization capacities of ForesightKV, we conduct experiments on three datasets: GPQA (Rein et al., 2023), LiveCodeBench (Jain et al., 2025), and Gov Report (Bai et al., 2024). The experimental results of Qwen3-4B are shown in Table 5. We can observe that our proposed ForesightKV achieve better performances on both long-text prefilling

(summarization) and generation (science and code generation) tasks. Since we only train the scoring models on math tasks, these results demonstrate the superior generalization capacities of our method on both long input and output tasks.

Table 5: Generalization capacities evaluations of Qwen3-4B on GPQA, LiveCodeBench, and Gov Report benchmarks.

| Model | Method | GPQA | | | LiveCodeBench | | Gov |
|-------|--------|------|------|------|------|------|------|
| | | 1K | 2K | 4K | 1K | 2K | 1K |
| | Full | | 54.6 | | | 63.4 | 29.4 |
| Qwen3-4B | H2O | 25.2 | 29.7 | 43.1 | 28.3 | 42.5 | 23.31 |
| | SnapKV | 16.9 | 34.8 | 49.1 | 27.0 | 48.1 | 26.0 |
| | R-KV | 23.0 | 40.0 | 50.7 | 34.4 | 52.0 | 26.2 |
| | F-KV(*w/o* RL) | 44.2 | 51.2 | 52.4 | **55.7** | **61.5** | 27.2 |
| | F-KV | **45.2** | **51.3** | **53.7** | **55.7** | 61.1 | **28.4** |

**Efficiency of ForesightKV.** To evaluate the efficiency of ForesightKV on long-context generation tasks, we set the KV budget to 1K, 2K, and 4K tokens and compare it with the full KV cache. We measure the maximum concurrent batch size and throughput on one A800 GPU with generation lengths of 8K, 16K, and 32K tokens. As shown in Table 6, applying ForesightKV can keep a fixed batch size and throughput under different generation lengths. Additionally, it substantially increases the feasible batch size during generation and leads to significant throughput improvements. For instance, ForesightKV with a budget of 1024 tokens achieves up to a $9.79\times$ throughput gain on long-context generation of 32K tokens.

Table 6: Efficiency evaluation of ForesightKV. "#MCB" denotes the maximum concurrent batch size.

| Method | 8K | | | 16K | | | 32K | | |
|--------|------|------------|-------|------|------------|-------|------|------------|-------|
| | #MCB | Throughput | Ratio | #MCB | Throughput | Ratio | #MCB | Throughput | Ratio |
| Full | 48 | 139.39 | 1.00× | 24 | 73.40 | 1.00× | 11 | 37.73 | 1.00× |
| ForesightKV-1K | 96 | 375.10 | 2.69× | 96 | 372.35 | 5.07× | 96 | 369.43 | 9.79× |
| ForesightKV-2K | 70 | 272.48 | 1.95× | 70 | 270.65 | 3.69× | 70 | 268.36 | 7.11× |
| ForesightKV-4K | 48 | 198.58 | 1.42× | 48 | 196.32 | 2.67× | 48 | 193.95 | 5.14× |
| ForesightKV-8K | - | - | - | 36 | 114.32 | 1.56× | 36 | 112.99 | 3.03× |

## 5 RELATED WORK

**KV Cache Eviction.** To mitigate the memory and computational overhead of the KV cache, which scales linearly with context length, various techniques have been proposed to enhance model efficiency, including KV cache eviction (Zhang et al., 2023), merging (Wang et al., 2024), dynamic loading (Tang et al., 2024), low-rank decomposition (Chang et al., 2024), and quantization (Hooper et al., 2024; Liu et al., 2024). Among these, KV cache eviction methods exploit the inherent sparsity of attention mechanisms to permanently discard less important KV pairs based on predefined rules. These methods typically rely on heuristic rules, *e.g.,* positions, attention scores, and representations of KV (Xiao et al., 2024; Zhang et al., 2023; Cai et al., 2025; Li et al., 2024; Wu et al., 2024; Goel et al., 2025). However, such heuristic-based approaches are often suboptimal for long-generation tasks. Additionally, some work uses trainable methods to discard unimportant caches (Lancucki et al., 2025; Zeng et al., 2024; Huang et al., 2024). These methods make a one-time judgment on the importance of the KV cache and then permanently discard it, which makes it difficult to capture the changing importance as the sequence evolves. Our method is the first to jointly use supervised and reinforcement learning to optimize the KV Cache eviction process. It dynamically captures the importance of KV pairs to decide on their eviction, leading to improved performance.

**Efficient Long-Context Generation.** With the extension of context windows and the enhancement of long-context abilities of LLMs, long-text generation tasks, particularly for long-reasoning

tasks, have emerged as a focal point of research. After reinforcement learning, the model accurately solves problems by generating a lengthy reasoning process (Yang et al., 2025a). To further improve efficiency, some studies have reduced the model's generation length by designing reinforcement learning algorithms (Cheng et al., 2025) and employing model merging (Wu et al., 2025). Other studies focus on reducing the computational cost associated with generating each token via speculative decoding (Yang et al., 2025b) or KV cache compression (Cai et al., 2025), without changing the generation lengths. Our work falls into the latter category, improving efficiency by reducing computational and storage overhead through KV cache eviction.

## 6 CONCLUSION

In this paper, we introduce ForesightKV, a novel two-stage training method for KV cache eviction tailored for long-context reasoning tasks. In the first stage, we design a Golden Eviction algorithm to obtain ideal eviction data, which is used to train scoring models with a Pairwise Ranking Loss. In the second stage, we model the eviction task as a Markov Decision Process and apply reinforcement learning to mitigate the significant performance loss on low-entropy tokens that can occur after eviction. Experiments on two reasoning benchmarks demonstrate that ForesightKV outperforms leading baselines with only half the cache budget. Ablation studies confirm that both the supervised and reinforcement learning stages are crucial to its success. We believe this reinforcement learning perspective opens up a promising new research direction for adaptive KV cache management in long-context reasoning.

## ETHICS STATEMENT

During the research process, we strictly adhered to academic standards and ethical guidelines. All the data used in our experiments strictly follows the ethical standards, *i.e.,* it contained no personal privacy information, no content that violates human values, and no biased or offensive material. Our research aims to enhance the intelligence of large language models, with the goal of enabling AI technologies to better assist all humankind, contributing to society and human well-being. We only use the large language models to examine and correct the grammar mistakes of our paper, and we manually review the content generated by the AI assistants to ensure the rigor and accuracy of our paper.

## REPRODUCIBILITY STATEMENT

To ensure the reproducibility of our work, we provide a detailed description of the algorithmic details in Section 3. Moreover, we present all the details about the implementation of our experiments in Section 4 and Appendix D, including the method for training data generation, hyperparameters of training and evaluation, and the details of the evaluation benchmarks and baselines. Furthermore, we also provide the code of our approach in the Supplementary Material. We believe the above information can help readers and researchers to reproduce our work.

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

## A LLM USAGE

We only use the large language model to examine and correct the grammatical mistakes in our paper.

## B MODEL ARCHITECTURES AND SAMPLING ALGORITHM

### B.1 MODEL DESIGN

In KV cache eviction, the design of a scoring model to accurately assess the importance of each key-value pair is crucial. To balance computational efficiency and accuracy, we use an MLP to score the importance of each KV pair. To more fully leverage existing information for prediction, we select two types of information as input:

- *Attention Features*: Inspired by H2O (Zhang et al., 2023) and SnapKV (Li et al., 2024), we use attention scores from both recent windows and cumulative history. For recent windows, we adopt sizes of 8, 16, and 32, along with a window size equal to the update length. For cumulative history, we aggregate attention scores across chunks using two variants: a direct sum and a decayed sum with a per-chunk decay factor of 0.9. For GQA, we concatenate the attention features of all heads within a group as the final attention feature $\mathbf{a}_n \in \mathbb{R}^{6g}$.

- *KV Representations*: We directly concatenate the hidden states of keys $\mathbf{k}_n \in \mathbb{R}^D$ and values $\mathbf{v}_n \in \mathbb{R}^D$ without additional transformation.

Thus, the input feature $\mathbf{x}_n = \text{Concat}([\mathbf{k}_n, \mathbf{v}_n, \mathbf{a}_n]) \in \mathbb{R}^{6g+2D}$. Subsequently, the input feature is sent into the MLP model $\pi_\phi$ to obtain the importance scores:

$$\phi_n = \pi_\theta(\mathbf{x}_n) = \sigma(\mathbf{x}_n \boldsymbol{W}_1 + \mathbf{b}_1)\boldsymbol{W}_2 + \mathbf{b}_2, \tag{11}$$

where $\boldsymbol{W}_1 \in \mathbb{R}^{(6g+2D)\times H}$, $\boldsymbol{W}_2 \in \mathbb{R}^{H\times 1}$, $\boldsymbol{b}_1 \in \mathbb{R}^H$, $\boldsymbol{b}_2 \in \mathbb{R}^1$, $H$ is the immediate size.

### B.2 SAMPLING ALGORITHM

After computing the importance score of each KV pair, we apply a Top-$K$ Multinomial Sampling strategy to determine which pairs to retain and which to evict. The procedure consists of two steps: (i) we first construct a candidate set by selecting the top-$K'$ pairs with the highest scores; (ii) we then normalize their scores with a softmax function and perform multinomial sampling to select $K$ pairs for eviction. As shown in Section 4.3, this approach outperforms both standard top-$K$ and multinomial sampling, while introducing stochasticity that encourages exploration, making it particularly suitable for reinforcement learning settings.

## C ANALYSIS OF LOW-ENTROPY TOKENS

We take samples from the reasoning traces and use Qwen3-4B with R-KV to predict the token at each position based on the maximum probability, and then compare these predictions with the original ones. We present the token changes of the top loss increase in low-entropy tokens in Table 7. We can observe that these tokens are often related to factual errors, *e.g.,* numbers and symbols. These errors may influence the following reasoning results.

Table 7: Error predictions of low-entropy tokens. The red and blue denote the predicted and original tokens in the sentence.

| | |
|---|---|
| Golden | $8 \times 9 \times 5 \times 7 = 2520$. |
| Prediction | $8 \times 9 \times 5 \times 7 = 8520$. |
| Golden | z = f(x)f(y) = (-1)(-1) = 1. |
| Prediction | z = f(x)f(y) + (-1)(-1) = 1. |

# D EXPERIMENTS DETAILS

## D.1 TRAINING SETUP

**Data Preparation.** We begin by preparing the training data using the Qwen3-4B model to generate reasoning traces on the STILL dataset (Min et al., 2024). The generation process employs the same settings as those used during the evaluation phase. From the generated outputs, we select only the corrected traces whose lengths exceed 4096 tokens. This curated collection of long, corrected reasoning traces serves as the dataset for the subsequent supervised training and reinforcement learning stages.

**Supervised Training** Following data preparation, we conduct supervised fine-tuning on the filtered dataset. The training is performed with a batch size of $8$ and a learning rate of $1e^{-2}$ for a total of $1000$ steps, utilizing a Cosine learning rate scheduler to anneal the learning rate. For our proposed method, we set the hyperparameter $m$ for the Pairwise Ranking Loss to $0.01$. The budget $B$ is configured to $1024$, and the eviction length $4$ is set to $2048$.

**Reinforcement Learning** In the reinforcement learning phase, we configure the training with a batch size and a mini-batch size of $32$. We employ a fixed learning rate of $3e^{-4}$ with a $10$ step of warmup and train the model for $200$ steps. For each sample in a batch, we generate $8$ eviction trajectories, and each batch of data is trained for one epoch without a mini-batch size. To balance efficiency and performance, the budget $B$ and eviction length $L$ are set to either $(1024, 256)$ or $(2048, 512)$, conditional on whether the sequence length is shorter or longer than 12K tokens. The KL penalty coefficient $\beta$ is set to $0.01$ for Qwen3-4B and $0.03$ for Qwen3-1.7B. Finally, the margin $\eta$ in the reward function is set to $1.5$ while $\epsilon$ is set to $0.2$.

## D.2 BASELINES

We set H2O (Zhang et al., 2023), SnapKV (Li et al., 2024), and R-KV (Cai et al., 2025) as baselines.

- *H2O*. H2O uses cumulative attention scores as the evaluation metric. At each step, the one with the highest cumulative attention score is retained. To maintain consistency, we also keep a budget of $B$ and perform pruning every $L$ tokens.
- *SnapKV*. SnapKV is a KV cache compression method for long inputs, which uses the last segment of queries as an observation window and determines importance based on their attention scores. Cai et al. (2025) transforms this into a compression method targeting long outputs, assessing the importance of the current step every $L$ tokens by utilizing the attention scores of the final 8 tokens.
- *R-KV*. Based on the aforementioned SnapKV, R-KV also takes into account the redundancy among tokens. It weighs the metrics of window attention score and token-to-token similarity to serve as the importance judgment for a KV pair. The weights for each are 0.1 and 0.9, respectively.

## D.3 REWARD CHANGES ACROSS TRAINING

We present the changes of loss $\mathcal{L}_{\text{ours}}$ with the different training steps. As shown in Figure 4, we can observe that during the initial training phase, the model exhibited unstable loss behavior, fluctuating relative to the original model. After a certain number of training steps, the loss consistently decreased, ultimately yielding better results. This indicates that the policy models can achieve effective exploration to improve the reasoning qualities.

## D.4 EXPERIMENTS WITH MORE MODELS

We evaluate the performance with DeepSeek-R1-Distill-Qwen-7B (DeepSeek-AI et al., 2025) on AIME2024, as shown in Table 8. Notably, we evaluate Duo-Attetnion on the compression ratio at 50% and 75%, which are compared with the budget of 2K and 4K. We can observe that our method can also achieve better performance than other methods.

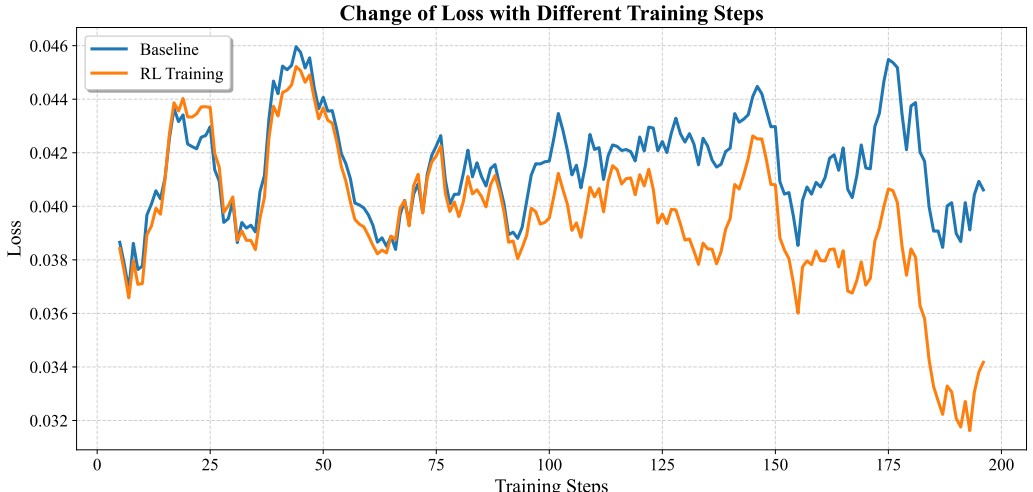

Figure 4: Change of losses with different training steps.

Furthermore, we extend our evaluation to MiniCPM-4.1-8B (Xiao et al., 2025) with a maximum sequence length of 32K, as presented in Table 9. The results demonstrate that our approach consistently outperforms alternative methods even with this larger model. However, due to the significantly longer generation lengths compared to the Qwen series models, we observe that more substantial KV cache budgets are necessary to maintain optimal performance.

Table 8: Performance evaluation of DeepSeek-R1-Distill-Qwen-7B on AIME 24 benchmark.

| Method | 1K | 2K | 4K |
|---|---|---|---|
| Full | | 52.6 | |
| SnapKV | 22.2 | 41.2 | 49.2 |
| H2O | 25.3 | 36.6 | 46 |
| R-KV | 26.8 | 43.1 | 50.5 |
| Duo-Attention | - | 3.3 | 13.3 |
| ForesightKV(\wo RL) | 43.1 | 50.0 | 54.2 |
| ForesightKV | 44.6 | 52.9 | 54.3 |

Table 9: Performance evaluation of MiniCPM-4.1-8B on AIME 24 benchmark.

| Method | 2K | 4K | 8K |
|---|---|---|---|
| Full | | 63.3 | |
| R-KV | 13.3 | 36.7 | 50.0 |
| SnapKV | 10.0 | 33.3 | 53.3 |
| ForesightKV(*w/o* RL) | 43.3 | 53.3 | 63.3 |

## D.5 PERFORMANCE ON LONGBENCH

We evaluate the generalization performance of our method with Qwen3-4B on LongBench, comparing it against SnapKV and H2O under a maximum generation length of 2K tokens[1]. As illustrated in Table eftab:longbench, our approach demonstrates superior performance on question-agnostic tasks

---

[1]R-KV encounters out-of-memory issues on several tasks.

that require leveraging information from the entire input, such as summarization and in-context learning. However, for question-aware tasks like question answering and retrieval, SnapKV performs better by utilizing the question to selectively evict KV pairs receiving less attention from the question. Trained on long-generation tasks, ForesightKV naturally prioritizes retaining KV pairs that contribute more substantially to all subsequent generations. Consequently, on long-input tasks, our method exhibits a more question-agnostic behavior, favoring the preservation of globally important key-value pairs.

Table 10: Performance evaluation of Qwen3-4B on LongBench.

|  | SingleQA | MultiQA | Syn | Summarization | ICL | Code |
|---|---|---|---|---|---|---|
| Full | 45.42 | 69.91 | 54.14 | 26.00 | 41.88 | 6.40 |
| ForesightKV | 40.09 | 50.47 | 30.11 | **25.12** | **40.44** | 6.76 |
| SnapKV | **42.46** | **60.92** | **41.64** | 23.95 | 37.82 | 6.40 |
| H2O | 33.48 | 29.16 | 9.56 | 21.82 | 27.89 | **8.37** |

## D.6 EXPERIMENTS WITH RPC

We compare our method with RPC (Song et al., 2025) on Qwen3-4B, as shown in Table 11. We can observe that our method outperforms RPC on AIME2024 under the same KV cache budgets. Specifically, RPC can be viewed as an example of SnapKV under our settings with different pooling method.

Table 11: Performance evaluation of Qwen3-4B on RPC.

| Method | 1K | 2K | 4K |
|---|---|---|---|
| RPC | 10.00 | 36.7 | 70.2 |
| ForesightKV | 54.5 | 70.2 | 69.2 |

