# OpenReview forum: "ForesightKV: Optimizing KV Cache Eviction for Reasoning Models by Learning Long-Term Contribution"
_ICLR.cc/2026/Conference — Submitted to ICLR 2026_

### Official Review · Reviewer_nPNm · 2025-10-21

**Soundness:** 3
**Presentation:** 3
**Contribution:** 3
**Rating:** 6
**Confidence:** 4

**Summary:**

The authors propose an effective KV cache compression method to address memory and computational challenges in long-sequence generation scenarios. The approach relies on a two-stage training framework to obtain a scorer model that can accurately predict the importance of KV cache entries. This method achieves significant performance improvements on the AIME24 and AIME25 datasets.

**Strengths:**

- The proposed two-stage training framework effectively addresses the KV Cache compression problem in long-sequence generation, resulting in improved accuracy.
- The achieved performance improvements are notable and consistently stable across the reported settings.

**Weaknesses:**

- The work lacks evaluation on larger models and architectures beyond the Qwen family, which is necessary to verify the consistency and generalizability of the results.
- The experimental evaluation is limited to AIME24 and AIME25. A broader range of tasks, e.g., question answering, code generation, and summarization, should be assessed to demonstrate robustness.

**Questions:**

- Is the trained scorer model used for head-wise selection of the KV cache tokens?
- Can the authors provide evidence or a dedicated experiment to demonstrate that the proposed method effectively solves the problem illustrated in Figure 1?

---

> ### Author Response · Authors · 2025-11-21
>
> Thank you for your helpful suggestions!
> ### W1: Lack of models beyond Qwen-3 family.
> We conducted experiments on DeepSeek-R1-Distill-Qwen-7B, and the results show that under the same budget, our method performs significantly better than other approaches on the AIME24 dataset. This demonstrates that our method can successfully transfer to other models.
> |                       | 1K   | 2K   | 4K   |
> |-----------------------|------|------|------|
> | Full                  |      | 52.6 |      |
> | SnapKV                | 22.2 | 41.2 | 49.2 |
> | H2O                   | 26.8 | 43.1 | 50.5 |
> | RKV                   | 25.3 | 36.6 | 46  |
> | ForesightKV (w.o. RL)   | 43.1 | 50  | 54.2 |
> | ForesightKV            | 44.6 | 52.9 | 54.3 |
>
> ### W2: Lack of datasets beyond AIME.
> To evaluate the generalization capacities of ForesightKV, we conduct experiments on three datasets: GPQA, LiveCodeBench, and Gov Report. The experimental results of Qwen3-4B are shown in the following Table. We can observe that our proposed ForesightKV achieves better performances on both **long-text prefilling (summarization) and generation (science and code generation) tasks**. Since we only train the scoring models on **math** tasks, these results demonstrate the **superior generalization capacities** of our method on both long input and output tasks.
>
> | Model        | Method        | GPQA 1K | GPQA 2K | GPQA 4K | LiveCodeBench 1K | LiveCodeBench 2K | Gov 1K |
> |--------------|---------------|---------|---------|---------|-------------------|-------------------|--------|
> | **Qwen3-4B** | Full          | —       | 54.6    | —       | 63.4             | 63.4             | 29.4   |
> |              | H2O           | 25.2    | 29.7    | 43.1    | 28.3             | 42.5             | 23.31  |
> |              | SnapKV        | 16.9    | 34.8    | 49.1    | 27.0             | 48.1             | 26.0   |
> |              | R-KV          | 23.0    | 40.0    | 50.7    | 34.4             | 52.0             | 26.2   |
> |              | ForesightKV (w/o RL) | 44.2    | 51.2    | 52.4    | **55.7**         | **61.5**         | 27.2   |
> |              | ForesightKV          | **45.2**| **51.3**| **53.7**| **55.7**         | 61.1             | **28.4** |
>
>
> ### Q1: Is the trained scorer model used for head-wise selection of the KV cache tokens?
> Yes. Current LLMs are often based on GQA, where attention heads within each group share a single KV Cache. Our approach is to design a dedicated scoring model for each KV Cache group. This model utilizes the information from the individual attention heads within that group, as well as the KV-pairs themselves, to determine the importance of each KV-pair. Subsequently, the KV Cache is managed based on these calculated importance scores.
>
>
> ### Q2: Can the authors provide evidence or a dedicated experiment to demonstrate that the proposed method effectively solves the problem illustrated in Figure 1?
>
> Because the number of KV pairs in long sequences is extremely large, the attention maps after masking are difficult to inspect directly. Therefore, we instead measure the similarity of outputs of attention heads before and after compression. We conducted experiments on Qwen3-4B while retaining a KV cache of size 1024 and 2048, and the results are shown below. We can see that the hidden-state similarity preserved by our method is higher than that of other approaches, indicating that our method is better at retaining the most critical tokens for different heads.
> |    | ForesightKV  | RKV    | SnapKV | H2O    |
> |----|--------------|--------|--------|--------|
> | 1K | **0.9736**       | 0.9628 | 0.9620 | 0.9711 |
> | 2K | **0.9889**       | 0.9782 | 0.9816 | 0.9845 |

---

> ### Comment · Area_Chair_1gvv · 2025-11-23
>
> Dear reviewer,
>
> Thanks for your time and effort in reviewing ICLR2026 submissions. The authors have submitted their responses to your review. Please take the time to read and raise your further comments, and discuss with the authors.
>
> Best regards,
>
> AC

---

### Official Review · Reviewer_ewCC · 2025-10-29

**Soundness:** 2
**Presentation:** 3
**Contribution:** 2
**Rating:** 4
**Confidence:** 4

**Summary:**

The authors introduce a training-based KV cache eviction framework to accurately predict KV cache importance, thereby enhancing the quality of cache compression. The method shows a notable advantage when applied to the Qwen-1.5B and Qwen-4B model.

**Strengths:**

- The experimental is well-designed, and the ablation studies are well-considered.
- The paper is easy to follow.

**Weaknesses:**

- In the efficiency evaluation, the authors compress sequence lengths from 16K and 32K to 1K and 2K, achieving substantial efficiency gains through extremely high compression ratios. Notably, even on AIME 24 and AIME 25, where the average length is under 16K, the method incurs significant accuracy drops when the KV budget is limited to 1K, with accuracy losses exceeding 40% in some cases. Thus, such aggressive compression may be impractical in real-world scenarios, limiting the practical applicability of the findings.

- The experiments were conducted only on very small-scale Qwen models with fewer than 4B parameters. The effectiveness of the proposed method on other architectures and larger-scale LLMs requires further validation.

- The evaluation datasets are overly limited. AIME 24 and AIME 25 contain only a small number of samples and are both restricted to mathematical reasoning tasks. Following common practice in this field, the authors should evaluate the effectiveness of their method on a broader range of tasks, such as those in LongBench.

- A comparison with other KV cache optimization methods (e.g., DuoAttention) is necessary to establish the relative merit of the proposed approach.

**Questions:**

- Can the authors report end-to-end speedup results across different datasets? Additionally, the authors should report throughput improvements under more practical compression ratios, such as compressing from 8K to 4K.

- What is the motivation for incorporating a pooling operation? Specifically, why is mean pooling followed by max pooling chosen in the design (Figure 2)?

- In Table 5, why are the MCB metrics identical for the Full Cache baseline and ForesightKV-2K at 16K generation length?

---

> ### Author Response · Authors · 2025-11-21
>
> Thank you for your helpful suggestions!
> ### W1: Efficiency evaluation is conducted on 1K and 2K budgets, which lead to significant performance drop.
> First, our experiments on the 1K and 2K budgets demonstrate the clear superiority of our method over others, showing that even under low budgets, our approach can largely preserve the model’s capabilities.
> Second, we also evaluated the efficiency of our method under a 4K budget, where we observed that its throughput increased from 126 to 198 and from 73 to 196 tokens per second under the 8K and 16K generation lengths, respectively. The results demonstrate the efficiency of our method during long generation tasks.
>
> ### W2: Lack of larger models beyond Qwen-3 family.
> We conducted experiments on DeepSeek-R1-Distill-Qwen-7B, and the results show that under the same budget, our method performs significantly better than other approaches on the AIME24 dataset. This demonstrates that our method can successfully transfer to other models.
> |                       | 1K   | 2K   | 4K   |
> |-----------------------|------|------|------|
> | Full                  |      | 52.6 |      |
> | SnapKV                | 22.2 | 41.2 | 49.2 |
> | H2O                   | 26.8 | 43.1 | 50.5 |
> | RKV                   | 25.3 | 36.6 | 46   |
> | ForesightKV (w.o. RL) | 43.1 | 50.0   | 54.2 |
> | ForesightKV           | 44.6 | 52.9 | 54.3 |
>
>
> ### W3: Lack of datasets beyond AIME.
> To evaluate the generalization capacities of ForesightKV, we conduct experiments on three datasets: GPQA, LiveCodeBench, and Gov Report. The experimental results of Qwen3-4B are shown in the following Table. We can observe that our proposed ForesightKV achieve better performances on both **long-text prefilling (summarization) and generation (science and code generation) tasks.** Since we only train the scoring models on **math** tasks, these results demonstrate the **superior generalization capacities** of our method on both long input and output tasks.
>
> | Model        | Method        | GPQA 1K | GPQA 2K | GPQA 4K | LiveCodeBench 1K | LiveCodeBench 2K | Gov 1K |
> |--------------|---------------|---------|---------|---------|-------------------|-------------------|--------|
> | **Qwen3-4B** | Full          | —       | 54.6    | —       | 63.4             | 63.4             | 29.4   |
> |              | H2O           | 25.2    | 29.7    | 43.1    | 28.3             | 42.5             | 23.31  |
> |              | SnapKV        | 16.9    | 34.8    | 49.1    | 27.0             | 48.1             | 26.0   |
> |              | R-KV          | 23.0    | 40.0    | 50.7    | 34.4             | 52.0             | 26.2   |
> |              | ForesightKV (w/o RL) | 44.2    | 51.2    | 52.4    | **55.7**         | **61.5**         | 27.2   |
> |              | ForesightKV          | **45.2**| **51.3**| **53.7**| **55.7**         | 61.1             | **28.4** |
>
> ### W4: Lack of comparison with DuoAttention
> We tested DuoAttention on DeepSeek-R1-Distill-Qwen-7B, and at compression ratios of 50% and 75%, the model achieved only 13.3 and 3.3 pass@1 on AIME24， which are lower than ForesightKV’s performance at 4K and 2K (which use even higher compression while remaining nearly lossless, as presented in W2). We hypothesize that, compared with long-text input tasks, long-text reasoning tasks rely more heavily on distant information across many heads, and the required operations are not limited to retrieval alone.

---

> > ### Author Response · Authors · 2025-11-21
> >
> > ### Q1: End-to-End Speedup across the whole dataset and throughput improvements under 8K to 4K.
> >
> > End-to-End Speedup: On DeepSeek-R1-Distill-Qwen-7B, we evaluated the whole cost under the batch size of 8, and each sample is tested 8 times. It takes 8 and 9.5 hours under the budget of 1K and 2K, and 17.5 hours for the full KV cache. Thus, even under the same budget, our method can achieve a about 2 times speedup.
> > Throughput improvements under 8K to 4K: we also evaluated the efficiency of our method under a 4K budget, where we observed that its throughput increased from 126 to 198 and from 73 to 196 tokens per second under the 8K and 16K generation lengths, respectively. The results demonstrate the efficiency of our method during long generation tasks.
> >
> > ### Q2: Motivation for pooling in Golden Eviction, and why mean pooling followed by max pooling?
> >
> > As discussed in our paper, we aim to determine whether each key–value pair should be retained based on its future importance. To do this, we need to pool future attention scores to estimate their prospective contribution. First, as identified in Section 2.2, many attention patterns exhibit block-wise characteristics, i.e., consecutive queries tend to assign similar attention weights to the same key. Therefore, we apply a block operation and use mean pooling within each block. Next, since the importance of a key–value pair changes dynamically over the sequence, we believe that if it plays a crucial role in any future block (even if it is not needed afterward), it should be preserved. Hence, we apply max pooling across blocks.
> >
> >
> > ### Q3: Why MCB of ForesightKV-2K is the same as the Full Cache?
> >
> > Thank you for your insightful observation. The significant memory overhead arises because we compute the attention matrix for the entire layer simultaneously when extracting attention features. In practice, we can mitigate peak memory usage by adopting a head-wise computation strategy. With this approach, at a sequence length of 2K, we were able to increase the MCB from 24 to 64, while simultaneously improving throughput from 208 to 272 tokens/s. Furthermore, we believe that further optimizing the attention computation during feature extraction (e.g., processing different samples within a batch individually) can lead to even higher MCB.

---

> > > ### Comment · Reviewer_ewCC · 2025-11-25
> > >
> > > Thank you for your response. However, I could not find many of the numbers you mentioned, and it seems you may have forgotten to upload the revised paper. The values in the tables are also difficult to follow.
> > >
> > > W1. At very low budgets (e.g., 1K), ForesightKV loses nearly one-third of its performance. I do not think such a compression is practically meaningful. In the paper, only compressing a 16K budget down to 4K preserves reasonable performance. Therefore, to make the efficiency experiments meaningful, you should at least evaluate compression from 32K to an 8K budget. In addition, I could not find the experiments under the 4K budget; you should explicitly indicate where these results come from.
> > >
> > > W2. Please note that my concern is that the evaluation is restricted to Qwen models; DeepSeek-R1-Distill-Qwen-7B is still part of the Qwen family. Robust evaluations of eviction methods typically test across multiple model families. I recommend that the authors follow this practice as well.
> > >
> > > W3. This part of your response does alleviate some of my concerns. However, it would be better to evaluate on a benchmark covering multiple predefined tasks, such as LongBench (which I mentioned), to rule out the possibility of cherry-picking datasets.
> > >
> > > W4. I am glad to see that ForesightKV outperforms DuoAttention. However, these results need to be properly organized into tables or figures and included in the main paper.
> > >
> > > Q1 & Q3. I believe the rebuttal should include data tables together with clear analysis. In its current form, it is difficult to grasp the main points or compare different cases. MCB are very important in practice. Your statement that “the significant memory overhead arises because we compute the attention matrix” is concerning. The authors should discuss and explain this issue in detail.

---

> ### Comment · Area_Chair_1gvv · 2025-11-23
>
> Dear reviewer,
>
> Thanks for your time and effort in reviewing ICLR2026 submissions. The authors have submitted their responses to your review. Please take the time to read and raise your further comments, and discuss with the authors.
>
> Best regards,
>
> AC

---

> ### Author Response · Authors · 2025-11-25
>
> Thank you for your suggestions! We have updated the PDF, and the revision is marked in Blue. We are preparing new experiments to further answer these questions
>
> * W1: We have updated the throughput evaluation in **Table 6**. Now, we are evaluating the throughput changes from a 32K to an 8K budget.
> * W2: We apologize for the misunderstanding; we are currently conducting additional experiments on other models. However, it is worth noting that Qwen-3 and DeepSeek-R1-Distill-Qwen already exhibit significant differences in both model architecture and training data distribution. This distinction, to a certain extent, demonstrates the generalizability of our method, as shown in **Appendix D.4 and Table 8.**
> * W3: First, we conducted experiments on GPQA, LiveCodeBench, and Gov report summarization tasks, as shown in **Table 5**. The primary motivation of this work is to demonstrate the capability of our method in handling long-output scenarios, which are demonstrated by the performance of GPQA and LiveCodeBench. However, to further demonstrate our generalization capabilities on long-input tasks, we are currently supplementing our evaluation with the full LongBench experiments
> * W4: We add the experiments of Duo-Attention and DeepSeek-R1-Distill-Qwen-7B in **Appendix D.4 and Table 8.**
> * Q1&Q3: In our implementation, we perform attention computation to extract specific attention features required for our scoring model. A naive computation of the entire layer simultaneously incurs a computational complexity of O(WLHN) per eviction step and results in substantial peak GPU memory usage, where W, L, H, and N denote the query length, KV context length, head dimension, and the total number of attention heads, respectively. To mitigate this, we optimized the process by performing attention computations sequentially for each head. This reduces the instantaneous computational overhead to O(WLH), thereby significantly lowering the peak memory footprint during inference. Thus, a large MCB can be ensured, as shown in **Table 6**.

---

> > ### Author Response · Authors · 2025-11-28
> > **Supplement Experiments**
> >
> > # W1: Efficiency Benchmarks
> > We have included additional speed benchmarks using an 8K budget with 16K and 32K context lengths. As demonstrated, the use of ForesightKV achieves speedups of 56% and 203%, respectively, compared to the original model, proving the significant efficiency advantages of our method. A comprehensive breakdown of all efficiency metrics is presented in **Table 6.**
> > # W2: Performance on MiniCPM-4.1-8B
> > We have supplemented our evaluation with results on MiniCPM-4.1-8B, testing on AIME2024 and tasks with 32K generation lengths. The results in **Table 9** indicate that our method outperforms other KV cache compression algorithms and achieves performance comparable to the original model at an 8K budget. It is worth noting that because this model typically exhibits longer generation lengths, a more conservative budget allocation is required.
> > # W3: LongBench Results and Scope Analysis
> > We have also provided results on LongBench, which can be found in **Table 10**. Specifically, our method achieves superior performance on question-agnostic tasks (Summarization, ICL), fully demonstrating its generalization capabilities. On question-aware tasks (Question answering, Retrieval), our performance is slightly lower than SnapKV. This is because SnapKV explicitly leverages attention from the final question segment for KV selection. In contrast, our method is specifically trained for and tailored to long-output tasks. Consequently, we prioritize retaining tokens that are likely to be important in the future (which are often more question-agnostic) rather than focusing solely on the immediate query. Therefore, a slight performance trade-off on question-aware tasks is expected.
> >
> > Finally, we emphasize that our method is designed for long-output generation, where it excels on benchmarks such as AIME, GPQA, and LiveCodeBench. This focus aligns with prior works in this domain, such as RKV, which also concentrated its testing on similar task types.

---

### Official Review · Reviewer_QH1s · 2025-10-30

**Soundness:** 2
**Presentation:** 3
**Contribution:** 2
**Rating:** 4
**Confidence:** 4

**Summary:**

This paper introduces ForesightKV, a two-stage KV cache eviction method for long-context generation. A lightweight scoring model is trained via 1) supervised pairwise ranking using future attention–based “Golden Eviction” labels and 2) GRPO-style reinforcement learning to reduce loss spikes on low-entropy tokens. Evaluations on Qwen3-1.7B and Qwen3-4B and AIME benchmarks show higher pass@1 under limited KV budgets and throughput gain.

**Strengths:**

1. Prior eviction work focuses on heuristics; this paper introduces a learned scoring policy.
2. Empirical attention visualizations provide qualitative motivation.
3. Parameter efficient training with smaller scorer, backbone LLM remains frozen.
4. Throughput and batch-size improvements are convincing.

**Weaknesses:**

1. All experiments focus on math reasoning (AIME), and the scorer is trained on reasoning traces from STILL-like data. This raises serious concerns about domain overfitting and limits claims of generality.
2. KV eviction is most relevant when serving >7B parameters. It is unclear whether attention patterns and eviction policies scale.
3. The RL reward specifically penalizes spikes on low-entropy symbolic tokens. This may not generalize to summarization, code.
4. Many practical workloads require reading long documents, not only generating long reasoning traces.
5. Throughput improves due to smaller active KV, but scoring overhead is not separately profiled.

**Questions:**

1. Isn’t the Golden Eviction label construction itself biased by the model being evaluated?
2. How does this method behave on long inputs instead of long outputs?
3. How does this interact with FlashAttention-style kernel fusion?
4. Why low-entropy tokens specifically? This may be task-domain specific (math, symbolic reasoning)
5. Do the learned policies generalize across models with different attention scaling behaviors (7B vs 70B)
6. Were the baseline methods tuned equivalently for long generation outputs? SnapKV was originally designed for long inputs. We should compare to some reasoning path compression baselines, such as Reasoning Path Compression (RPC) https://arxiv.org/abs/2505.13866

---

> ### Author Response · Authors · 2025-11-21
>
> Thank you for your insightful suggestions!
> ### W1： Lack of datasets beyond AIME.
> To evaluate the generalization capacities of ForesightKV, we conduct experiments on three datasets: GPQA, LiveCodeBench, and Gov Report. The experimental results of Qwen3-4B are shown in the following Table. We can observe that our proposed ForesightKV achieves better performances on both **long-text prefilling (summarization) and generation (science and code generation) tasks**. Since we only train the scoring models on **math** tasks, these results demonstrate the **superior generalization capacities** of our method on both long input and output tasks.
>
> | Model        | Method        | GPQA 1K | GPQA 2K | GPQA 4K | LiveCodeBench 1K | LiveCodeBench 2K | Gov Report 1K |
> |--------------|---------------|---------|---------|---------|-------------------|-------------------|--------|
> | **Qwen3-4B** | Full          | —       | 54.6    | —       | 63.4             | 63.4             | 29.4   |
> |              | H2O           | 25.2    | 29.7    | 43.1    | 28.3             | 42.5             | 23.3  |
> |              | SnapKV        | 16.9    | 34.8    | 49.1    | 27.0             | 48.1             | 26.0   |
> |              | R-KV          | 23.0    | 40.0    | 50.7    | 34.4             | 52.0             | 26.2   |
> |              | ForesightKV (w/o RL) | 44.2    | 51.2    | 52.4    | **55.7**         | **61.5**         | 27.2   |
> |              | ForesightKV          | **45.2**| **51.3**| **53.7**| **55.7**         | 61.1             | **28.4** |
> ### W2: Lack of experiments with large models
> We conducted experiments on DeepSeek-R1-Distill-Qwen-7B, and the results show that under the same budget, our method performs significantly better than other approaches on the AIME24 dataset. This demonstrates that our method can successfully transfer to other models.
> |                       | 1K   | 2K   | 4K   |
> |-----------------------|------|------|------|
> | Full                  |      | 52.6 |      |
> | SnapKV                | 22.2 | 41.2 | 49.2 |
> | H2O                   | 26.8 | 43.1 | 50.5 |
> | RKV                   | 25.3 | 36.6 | 46   |
> | ForesightKV (w.o. RL) | 43.1 | 50   | 54.2 |
> | ForesightKV           | 44.6 | 52.9 | 54.3 |
>
>
> ### W3: RL reward of low-entropy tokens beyond symbolic tasks
> We evaluate the changes in low-entropy tokens in code generation and summarization tasks with and without cache eviction. We observe that low-entropy tokens vary much more sharply than high-entropy tokens. In code generation, the number of low-entropy tokens increases by 75%, whereas high-entropy tokens increase by only 1%. In summarization, low-entropy tokens increase by 187%, while high-entropy tokens increase by 142%.
> |      | Math  | Code | Sum   |
> |------|-------|------|-------|
> | Low  | +147% | +75% | +187% |
> | High | +53%  | +1%  | +142% |
>
> In addition, across both of these task types, as well as in mathematical reasoning, symbol errors account for only part of the issues, and some textual descriptions also exhibit errors.
>
> Furthermore, our method serves as a universal framework that supports different rewards during the RL stage. Task-specific reward designs can be seamlessly integrated within this framework.
>
> ### W4: Processing long input is important, but is not evaluated
> First, we conducted experiments on long-input scenarios (Gov Report Summarization). The results in W1 show that even though our method was tuned on math tasks, it demonstrates strong generalization ability: with only a 1K budget, it outperforms other KV cache eviction methods.
>
> Second, most prior work has focused on optimizing long-input settings, while research on long-output scenarios remains relatively limited. Our work provides a more thorough exploration of how to better balance efficiency and performance in long-output generation.
>
> ### W5: Lack of a separate overhead of the KV cache management.
> We evaluate our method on Qwen3-4B with a budget of 2048. Specifically, we generate sequences of length 32K with a batch size of 64, which takes a total of 5813 seconds. Among this, our KV cache eviction mechanism accounts for only 157 seconds, representing just **2.7%** of the total time, which is effectively negligible. Moreover, due to the larger degree of parallelism, reduced attention computation, and lower cache storage overhead, our method achieves significantly higher computational efficiency, which is sufficient to cover the overhead of the scoring model.

---

> ### Author Response · Authors · 2025-11-21
>
> ### Q1: Isn’t the Golden Eviction label construction itself biased by the model being evaluated?
> Yes, the Golden Eviction metric inherently depends on the model itself. Golden Eviction provides an eviction trajectory tailored for each group of attention heads in a model’s GQA structure, rather than a universal trajectory applicable to all models or all sequences. Because different heads exhibit different KV characteristics and attention patterns, their optimal eviction strategies vary accordingly. Therefore, we must design a dedicated eviction process for each attention-head group to ensure the best performance.
> ### Q2: Performance on long-context input tasks.
> Our method is designed specifically for long-input scenarios, where it is crucial to capture the dynamically changing importance of tokens during future generation. Moreover, our approach can be applied in long-input settings. We directly evaluated Qwen3-4B together with the scoring models previously trained on mathematical tasks on the Gov Report summarization benchmark. As shown in W1, our method achieves better performance than prior approaches under the 1K budget (28.4 v.s. 26.2).
> ### Q3: Interaction with Flash-Attention
> We employ FlashAttention2 for attention computation during both the prefilling and decoding stages. Additionally, at each eviction step, we compute the attention scores by directly multiplying the query and key matrices. Since the sequence length involved in this computation is short, this step introduces only a minimal computational overhead.
> ### Q4: Specialness of low-entropy tokens and domain-generalization of these tokens.
> ●Specialness of low entropy tokens: As discussed in Section 2.2 of our paper, low-entropy tokens are typically associated with factual and detailed content [1]. Prior work has also shown that these tokens are closely tied to the model’s retrieval ability, as they often attend directly to earlier content [2]. Therefore, KV cache eviction,which harm the retrieval of previous content, tends to disproportionately damage the representations of these low-entropy tokens.
>
> ●Domain-specification of low entropy tokens: In addition, the loss of low-entropy tokens also tends to increase large on summarization and code generation tasks. We evaluate the changes in low-entropy tokens in code generation and summarization tasks with and without cache eviction, and we observe that low-entropy tokens change much more sharply than high-entropy tokens. In code generation, low-entropy tokens increase by 75%, whereas high-entropy tokens increase by only 1%. In summarization, low-entropy tokens increase by 142%, while high-entropy tokens increase by 187%.
> |      | Math  | Code | Sum   |
> |------|-------|------|-------|
> | Low  | +147% | +75% | +187% |
> | High | +53%  | +1%  | +142% |
> ### Q5: Generalization of learned policies to different models
> ●First, our method can be applied to other models as well, but it requires training.  Since different attention heads have different inputs and patterns, the scoring models need to be optimized specifically for attention heads in each model.
>
> ●In addition, we also conducted experiments on DeepSeek-R1-Distill-Qwen-7B, and the results show that our method achieves better performance than other approaches
>
> ### Q6: Were all baselines tuned equivalently for long generation and comparison with RPC.
> ●Equivalent of baselines: First, we refer to R-KV, a KV cache eviction algorithm designed for long-output scenarios in reasoning models. This method modifies SnapKV by performing compression using the most recent window at fixed intervals, allowing it to better support long contexts. Other baselines in our paper, H2O and R-KV, are inherently designed for long-output settings. Therefore, all of our baselines are fully oriented toward long-output scenarios.
>
> ●Comparison to RPC: In addition, we include experiments with the RPC method, using the same budget and window-size configuration as the other baselines. As shown in the following table, on Qwen3-4B, our method still significantly outperforms RPC on AIME-2024.
> |    | RPC  | ForesightKV |
> |----|------|-------------|
> | 1K | 10   | 54.5        |
> | 2K | 36.7 | 70.2        |
> | 4K | 59.2 | 69.2        |
>
> [1] Beyond the 80/20 Rule: High-Entropy Minority Tokens Drive Effective Reinforcement Learning for LLM Reasoning. NeurIPS 2025.
> [2] Entropy-Based Decoding for Retrieval-Augmented Large Language Models. NAACL 2025.

---

> ### Comment · Area_Chair_1gvv · 2025-11-23
>
> Dear reviewer,
>
> Thanks for your time and effort in reviewing ICLR2026 submissions. The authors have submitted their responses to your review. Please take the time to read and raise your further comments, and discuss with the authors.
>
> Best regards,
>
> AC

---

### Official Review · Reviewer_nXgV · 2025-11-01

**Soundness:** 3
**Presentation:** 2
**Contribution:** 3
**Rating:** 4
**Confidence:** 4

**Summary:**

The idea of learning a KV-eviction policy is good, and the motivation for handling long context reasoning with dynamic KV importance is clear. The basic intuition that attention patterns shift and some tokens become important later makes sense, and the results on AIME show improvement, and the performance gains look reasonable.

However, the work is tied to math reasoning. The reward design and the low-entropy token focus seem to rely on math traces, where errors are clean and easy to detect. I was curious whether the main reason AIME is used is to make the system easier to train. If so, it raises questions about how well the method generalizes to other tasks.

Because this is a training-based method, transparency on the process matters, but many training details are not clearly explained. It is not obvious how many examples were used, how the “correct long reasoning trace” filtering works, how expensive the Golden Eviction step is, or how much tuning was needed to stabilize the RL phase. These steps look non-trivial, and right now it is hard to know how reproducible or practical the full pipeline is.

Overall, the results are promising for math reasoning, and the high-level idea is interesting. It would help the paper if it clarified the training pipeline more thoroughly and discussed how the method might extend beyond math, or what would be required to adapt the reward and supervision signals to other domains.

**Strengths:**

Clear motivation, meaningful gains on math reasoning, smart use of future attention and RL, and a promising results for learned memory over heuristic KV strategies.

**Weaknesses:**

Heavy reliance on math-specific reward signals, unclear generalization to other tasks, and missing training details that limit reproducibility and practical adoption.

**Questions:**

1. How does the approach generalize beyond math reasoning tasks where correctness and low-entropy signals are easy to define?

2. What would the reward function look like for dialogue, long-form QA, or code generation, where “important tokens” are harder to identify?

3. How many training samples and reasoning traces were used, and how were they filtered in practice?

4. What is the exact cost and procedure for generating the “Golden Eviction” oracle labels at scale?

5. How stable is the RL training stage, and what tuning was required to make it work?

6. Does the method still provide meaningful gains when applied to diverse long-context tasks rather than only AIME-style benchmarks?

7. What is the inference-time overhead of running the scorer network, and how does it compare to simpler heuristics at scale?

8. Could the scoring policy be integrated into the base model during training rather than as an external module?

9. If the method is applied to models with different architectures or training distributions, does the eviction policy still transfer?

---

> ### Author Response · Authors · 2025-11-21
>
> Thank you for your insightful suggestions!
> ### W1: Heavy reliance on math-specific reward signals
> First, our method does not rely on the math-specific reward signals commonly used in traditional large-model RL. Instead, our reward is designed by comparing the change in LM loss before and after cache compression along the sequence, which in principle, makes it applicable to any task.
>
> Secondly, we also verify that similar sharp increases in low-entropy tokens occur in both code generation and summarization tasks.  In code generation, the number of low-entropy tokens increases by 75%, whereas high-entropy tokens increase by only 1%. In summarization, low-entropy tokens increase by 142%, while high-entropy tokens increase by 187%.
>
> |      | Math  | Code | Sum   |
> |------|-------|------|-------|
> | Low  | +147% | +75% | +187% |
> | High | +53%  | +1%  | +142% |
> ### W2: Unclear generalization to other tasks
> To evaluate the generalization capacities of ForesightKV, we conduct experiments on three datasets: GPQA, LiveCodeBench, and Gov Report. The experimental results of Qwen3-4B are shown in the following Table. We can observe that our proposed ForesightKV achieves better performances on both **long-text prefilling (summarization) and generation (science and code generation) tasks**. Since we only train the scoring models on **math** tasks, these results demonstrate the **superior generalization capacities **of our method on both long input and output tasks.
>
> | Model        | Method        | GPQA 1K | GPQA 2K | GPQA 4K | LiveCodeBench 1K | LiveCodeBench 2K | Gov 1K |
> |--------------|---------------|---------|---------|---------|-------------------|-------------------|--------|
> | **Qwen3-4B** | Full          | —       | 54.6    | —       | 63.4             | 63.4             | 29.4   |
> |              | H2O           | 25.2    | 29.7    | 43.1    | 28.3             | 42.5             | 23.31  |
> |              | SnapKV        | 16.9    | 34.8    | 49.1    | 27.0             | 48.1             | 26.0   |
> |              | R-KV          | 23.0    | 40.0    | 50.7    | 34.4             | 52.0             | 26.2   |
> |              | ForesightKV (w/o RL) | 44.2    | 51.2    | 52.4    | **55.7**         | **61.5**         | 27.2   |
> |              | ForesightKV          | **45.2**| **51.3**| **53.7**| **55.7**         | 61.1             | **28.4** |
> ### W3: Missing training details that limit reproducibility and practical adoption.
> All of our training details and hyperparameter settings are provided in Appendix D.1, and we include the training code in the supplementary materials. We hope these resources will facilitate the reproduction of our results.
> Question
>
> ### Q1: Generalization beyond math tasks where correctness and low-entropy signals are easy to define.
>
> ●First, the low-entropy signals are universal on different tasks, which are presented in W1.
>
> ●Additionally, we provide the generalization experiments on W2. We can observe that though our method is trained on math tasks, the performance on other tasks is also better than baselines.
>
> ### Q2: Reward function on tasks where important tokens are harder to identify.
>
> First, our reward function is designed based on the changes in entropy and loss before and after compression, and therefore does not require identifying any special tokens.
>
> Second, in W1, we also verified the widespread presence of this low-entropy pattern and its strong connection to retrieval ability in long-context settings. Thus, we believe that our reward function can generalize to such tasks as well.
> ### Q3: Number of training samples and filter method.
> For the SFT and RL stages, we used 4000 and 6400 training samples, respectively. To construct these training datasets, we took the math questions from the STILL dataset and generated trajectories using Qwen3-4B. We selected trajectories longer than 4096 tokens whose final answers were correct. The details are provided in Appendix D.1.

---

> ### Author Response · Authors · 2025-11-21
>
> ### Q4: Cost and procedure for generating Golden Eviction labels.
> Procedure: As defined in Section 3.2 of our paper, golden eviction aims to remove, at each step, the KV pair with the smallest impact on future computations. Specifically, we divide each token’s future attention scores into blocks and average them to obtain block scores, then discard the KV pair whose maximum block score is the smallest. We provide the code implementation of golden eviction in the supplementary material.
>
> Cost: In terms of cost, our tests on a single A800 GPU show that constructing the golden eviction mapping for each sample takes only about 6 seconds. Notably, this entire process can be performed offline and does not affect training speed.
>
> ### Q5: Stability of RL training and what tuning are required.
> Overall, our RL training behaves similarly to standard RL training for LLMs: it fluctuates around the baseline during the early stages and begins to show a stable improvement in reward over the baseline in later stages, as illustrated in the appendix. The training hyperparameters are presented in Appendix D.1. In general, our method is quite robust to hyperparameter choices and requires no special tuning. The only component we adjust is the KL-divergence penalty (0.01 to 0.05) to maintain training stability. We also provide our training code in the supplementary material.
> ### Q6: Generalization to tasks beyond AIME
> To evaluate the generalization capacities of ForesightKV, we conduct experiments on three datasets: GPQA, LiveCodeBench, and Gov Report. The experimental results of Qwen3-4B are shown in the following Table. We can observe that our proposed ForesightKV achieve better performances on both **long-text prefilling (summarization) and generation (science and code generation)** tasks. Since we only train the scoring models on **math** tasks, these results demonstrate the **superior generalization capacities** of our method on both long input and output tasks.
>
> | Model        | Method        | GPQA 1K | GPQA 2K | GPQA 4K | LiveCodeBench 1K | LiveCodeBench 2K | Gov 1K |
> |--------------|---------------|---------|---------|---------|-------------------|-------------------|--------|
> | **Qwen3-4B** | Full          | —       | 54.6    | —       | 63.4             | 63.4             | 29.4   |
> |              | H2O           | 25.2    | 29.7    | 43.1    | 28.3             | 42.5             | 23.31  |
> |              | SnapKV        | 16.9    | 34.8    | 49.1    | 27.0             | 48.1             | 26.0   |
> |              | R-KV          | 23.0    | 40.0    | 50.7    | 34.4             | 52.0             | 26.2   |
> |              | ForesightKV (w/o RL) | 44.2    | 51.2    | 52.4    | **55.7**         | **61.5**         | 27.2   |
> |              | ForesightKV          | **45.2**| **51.3**| **53.7**| **55.7**         | 61.1             | **28.4** |
> ### Q7: Inferece overhead of scorer networks
> We evaluate our method on Qwen3-4B with a budget of 2048. Specifically, we generate sequences of length 32K with a batch size of 64, which takes a total of 5813 seconds. Among this, our KV cache eviction mechanism accounts for only 157 seconds, representing just **2.7%** of the total time, which is effectively negligible.
>
> In contrast, the computation of R-KV takes about 8.1% of the total time, which is much larger.
>
> Moreover, due to the larger degree of parallelism, reduced attention computation, and lower cache storage overhead, our method achieves significantly higher computational efficiency, which is sufficient to cover the overhead of the scoring model.
>
> ### Q8: Can the scoring policy be integrated into the base model
>
> That is possible, but doing so would require propagating gradients through the entire model in order to update the scoring models. Compared with our current strategy, which updates parameters solely based on the advantages, this would introduce significantly higher training overhead. Therefore, from a cost-efficiency perspective, we chose to implement the scoring model as an external module rather than integrating it into the main model.
>
> In addition, under the current framework, we could also use the scoring model to update the model’s parameters, although this would incur even higher memory (GPU) overhead.
>
> ### Q9: Can the eviction policy transfer to other models?
> Yes, our method can be applied to train different models, as demonstrated by our experiments. However, these methods require re-tuning and cannot be directly transferred. This necessity arises because our scoring model relies on features derived from the attention heads and the KV Cache as input. Similarly, the trajectory dropping mechanism also depends on the features of the attention heads. Consequently, retraining the scoring models is required for each application.

---

> ### Comment · Area_Chair_1gvv · 2025-11-23
>
> Dear reviewer,
>
> Thanks for your time and effort in reviewing ICLR2026 submissions. The authors have submitted their responses to your review. Please take the time to read and raise your further comments, and discuss with the authors.
>
> Best regards,
>
> AC

---

### Author Response · Authors · 2025-12-01
**General summary of our rebuttal**

Thank you to all reviewers and the Area Chair for your efforts. In response, we have provided extensive details in our rebuttal, which we believe will fully address the reviewers' concerns. Below, we provide responses to the main concerns raised:

## Insufficient model diversity

In **Table 8 and Table 9**, we report results on **DeepSeek-R1-Distill-Qwen-7B and  MiniCPM-4.1-8B**, showing that our method consistently outperforms all baselines on the AIME2024 task across all budgets. This demonstrates the model generality of our approach.

## Insufficient evaluation coverage

We further evaluated our method on long-output tasks such as GPQA and LiveCodeBench (as shown in **Table 5**), and it continues to achieve leading performance. Since our training was conducted only on mathematical tasks, these results strongly support the generalization ability of our method.

On long-input benchmarks such as LongBench (as shown in **Table 10)**, our method achieves the best results on question-independent tasks, while being slightly weaker than SnapKV on question-dependent tasks, because SnapKV explicitly uses the question to select KV pairs. This suggests that our approach primarily learns a task-agnostic notion of importance from mathematical training.

##  Efficiency

We evaluate efficiency across different budgets and output lengths in **Table 6**, and observe consistent speed improvements whenever the budget is smaller than the output length. In the end-to-end evaluation, our method also achieves faster overall inference.

Furthermore, we have updated the calculation logic for our attention features from a layer-wise to a head-wise approach. This significantly reduced peak video memory usage and increased the MCB.

We also measured the time overhead of our KV Cache Eviction phase, which accounts for only $0.027$ of the total computation time, rendering it effectively negligible.


## Transferability of Scoring Models
Our method is specifically designed for each group of heads in Grouped Query Attention (GQA). Since the input and attention features differ across heads, the optimal eviction trajectory naturally varies. Consequently, these scoring models must be obtained through training. Furthermore, we have demonstrated that our method can be successfully trained on different base models, exhibiting strong generalization capabilities.

## Generalization to Long-Input Tasks

 First, it is important to note that our method is primarily designed for long-output (generation) tasks. Following previous works (e.g., R-KV [1], RPC [2]), we conducted our testing on long-output scenarios. Therefore, long-input tasks are considered out-of-domain for this specific scope.

However, to address this, we provided generalization results on LongBench using the model trained on math tasks. As shown, our method performs better on question-agnostic tasks (such as Summarization and ICL), which fully demonstrates the generalizability of our approach.

## Comparison with Additional Baselines
We have compared our method with RPC and Duo-Attention, and the results are presented in **Table 11 and Table 9**, respectively. The data shows that our method achieves superior performance on the AIME2024 benchmark compared to these baselines.

## Hyperparameters and Data Settings
We have detailed our hyperparameter settings and data design in Appendix D.1. Additionally, we have provided the source code in the supplementary material for reproduction.

## Generalization of Low-Entropy Tokens
We analyzed the behavior of low-entropy and high-entropy tokens after applying Cache Eviction across math, code, and summarization tasks, as shown in **Table 1**. The results indicate that the loss increase for low-entropy tokens is significantly more pronounced compared to that of high-entropy tokens.

[1] R-KV: Redundancy-aware KV Cache Compression for Reasoning Models

[2] Reasoning Path Compression: Compressing Generation Trajectories for Efficient LLM Reasoning

---

### Meta-Review · Area_Chair_578E · 2026-01-11

**Summary:**

This paper proposes a method for training a scoring model (also utilizing RL) to measure KV importance in long-context reasoning. At inference, the KV eviction is conducted using this scoring model. Experiments were primarily conducted in the math domain, demonstrating improved KV compression capability compared to existing methods.

Most reviewers raised concerns about the method's generalizability and limited scale. While the authors provided additional experiments during the rebuttal period, these are more critical than those in the initially proposed paper. Given the limited rebuttal and discussion period, they cannot be fully resolved since more time would be needed to verify and review them. In the AC's judgment, even considering these issues, the methodology is not compelling enough to flip the decision to acceptance of the reviewers since it is already complicated (i.e., two-staged framework) with additional overhead. Given these considerations, the AC's decision is to reject this paper.

**Reviewer Concerns:**

The common concerns of reviewers can be summarized as follows:
- Generalizability to other domains (math-specific design and experiments)
- Small scale (Qwen3-1.7B, Qwen3-4B)
- Missing details (profiling overhead of scoring model) and comparisons

Additional experiments were provided to address these concerns, including experiments on additional datasets, such as GPQA and LiveCodeBench, additional models, such as DeepSeek-R1-Distill-Qwen-7B and MiniCPM-4.1-8B, and a comparison with DuoAttention. While these additions partially address the concerns, the paper still remains borderline from the reviewers' perspective. The methodology is not compelling enough to flip from the initial decision to acceptance since it is a two-stage framework with additional overhead, and the additional experiments should provide a more thorough analysis. The AC recommends submitting this paper with the additional experiments and analyses at the next venue.

**Reviewer Scores:**

As mentioned in the previous section, the AC finds that the reviewers are unlikely to change their initial scores.

---

### Decision · Program_Chairs · 2026-01-26

Reject